# An Active Learning Framework using Sparse-Graph Codes for Sparse Polynomials and Graph Sketching

**Xiao Li**
UC Berkeley
xiaoli@berkeley.edu

**Kannan Ramchandran**∗
UC Berkeley
kannanr@berkeley.edu

## Abstract

Let $f : \{-1, 1\}^n \to \mathbb{R}$ be an $n$-variate polynomial consisting of $2^n$ monomials, in which only $s \ll 2^n$ coefficients are non-zero. The goal is to learn the polynomial by querying the values of $f$. We introduce an active learning framework that is associated with a low query cost and computational runtime. The significant savings are enabled by leveraging sampling strategies based on modern coding theory, specifically, the design and analysis of *sparse-graph codes*, such as Low-Density-Parity-Check (LDPC) codes, which represent the state-of-the-art of modern packet communications. More significantly, we show how this design perspective leads to exciting, and to the best of our knowledge, largely unexplored intellectual connections between learning and coding.

The key is to relax the worst-case assumption with an ensemble-average setting, where the polynomial is assumed to be drawn uniformly at random from the ensemble of all polynomials (of a given size $n$ and sparsity $s$). Our framework succeeds with high probability with respect to the polynomial ensemble with sparsity up to $s = O(2^{\delta n})$ for any $\delta \in (0, 1)$, where $f$ is exactly learned using $O(ns)$ queries in time $O(ns \log s)$, even if the queries are perturbed by Gaussian noise. We further apply the proposed framework to graph sketching, which is the problem of inferring sparse graphs by querying graph cuts. By writing the cut function as a polynomial and exploiting the graph structure, we propose a sketching algorithm to learn the an arbitrary $n$-node unknown graph using only few cut queries, which scales *almost linearly* in the number of edges and *sub-linearly* in the graph size $n$. Experiments on real datasets show significant reductions in the runtime and query complexity compared with competitive schemes.

## 1 Introduction

One of the central problems in computational learning theory is the efficient learning of polynomials $f(\mathbf{x}) : \mathbf{x} \in \{-1, 1\}^n \to \mathbb{R}$. The task of learning an $s$-sparse polynomial $f$ has been studied extensively in the literature, often in the context of Fourier analysis for *pseudo-boolean functions* (a real-valued function defined on a set of binary variables). Many concept classes, such as $\omega(1)$-juntas, polynomial-sized circuits, decision trees and disjunctive normative form (DNF) formulas, have been proven very difficult [1] to learn in the worst-case with random examples. Almost all existing efficient algorithms are based on the *membership query* model [1, 6–8, 10, 11, 17], which provides arbitrary access to the value of $f(\mathbf{x})$ given any $\mathbf{x} \in \{-1, 1\}^n$. This makes a richer set of concept classes learnable in polynomial time $\text{poly}(s, n)$. This is a form of what is now popularly referred to as *active learning*, which makes queries using different sampling strategies. For instance, [3, 10] use regular subsampling and [9, 14, 18] use random sampling based on compressed sensing. However, they remain difficult to scale computationally, especially for large $s$ and $n$.

---

∗This work was supported by grant NSF CCF EAGER 1439725.

In this paper, we are interested in learning polynomials with $s = O(2^{\delta n})$ for some $\delta \in (0, 1)$. Although this regime is not typically considered in the literature, we show that by relaxing the "worst-case" mindset to an ensemble-average setting (explained later), we can handle this more challenging regime and reduce both the number of queries and the runtime complexity, even if the queries are corrupted by Gaussian noise. In the spirit of active learning, we design a sampling strategy that makes queries to $f$ based on modern coding theory and signal processing. The queries are formed by "strategically" subsampling the input to induce *aliasing* patterns in the dual domain based on sparse-graph codes. Then, our framework exploits the aliasing pattern (code structure) to reconstruct $f$ by peeling the sparse coefficients with an iterative simple peeling decoder. Through a coding-theoretic lens, our algorithm achieves a low query complexity (capacity-approaching codes) and low computational complexity (peeling decoding).

Further, we apply our proposed framework to *graph sketching*, which is the problem of inferring hidden sparse graphs with $n$ nodes by actively querying graph cuts (see Fig. 1). Motivated by bioinformatics applications [2], learning hidden graphs from additive or cross-additive queries (i.e. edge counts within a set or across two sets) has gained considerable interest. This problem closely pertains to our learning framework because the cut function of any graph can be written as a sparse polynomial with respect to the binary variables $\mathbf{x} \in \{-1, +1\}^n$ indicating a graph partition for the cut [18]. Given query access to the cut value for an arbitrary partition of the graph, how many cut queries are needed to infer the hidden graph structure? What is the runtime for such inference?

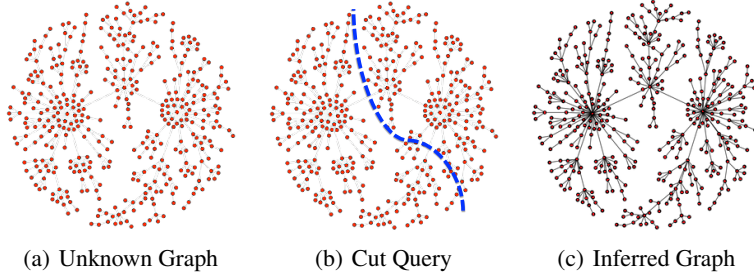

(a) Unknown Graph      (b) Cut Query      (c) Inferred Graph

Figure 1: Given a set of $n$ nodes, infer the graph structure by querying graph cuts.

Most existing algorithms that achieve the optimal query cost for graph sketching (see [13]) are *non-constructive*, except for a few algorithms [4, 5, 9, 18] that run in *polynomial* time in the graph size $n$. Inspired by our active learning framework, we derive a sketching algorithm associated with a query cost and runtime that are both *sub-linear in the graph size $n$* and *almost-linear in the number of edges*. To the best of our knowledge, this is the first constructive non-adaptive sketching scheme with sub-linear costs in the graph size $n$. In the following, we introduce the problem setup, our learning model, and summarize our contributions.

## 1.1 Problem Setup

Our goal is to learn the following polynomial in terms of its coefficients:

$$f(\mathbf{x}) = \sum_{\mathbf{k} \in \mathbb{F}_2^n} \alpha[\mathbf{k}] \chi_{\mathbf{k}}(\mathbf{x}), \ \forall \, \mathbf{x} \in \{-1, 1\}^n, \quad \mathbb{F}_2 := \{0, 1\}, \tag{1}$$

where $\mathbf{k} := [k[1], \cdots, k[n]]^T \in \mathbb{F}_2^n$ is the index of the monomial[1] $\chi_{\mathbf{k}}(\mathbf{x}) = \prod_{i \in [n]} x_i^{k[i]}$, and $\alpha[\mathbf{k}] \in \mathbb{R}$ is the coefficient. In this work, we consider an ensemble-average setting for learning.

**Definition 1** (Polynomial Ensemble). *The polynomial ensemble $\mathcal{F}(s, n, \mathcal{A})$ is a collection of polynomials $f : \{-1, 1\}^n \to \mathbb{R}$ satisfying the following conditions:*

- *the vector $\boldsymbol{\alpha} := [\cdots, \alpha[\mathbf{k}], \cdots]^T$ is $s$-sparse with $s = O(2^{\delta n})$ for some $0 < \delta < 1$;*

- *the support $\mathrm{supp}(\boldsymbol{\alpha}) := \{\mathbf{k} : \alpha[\mathbf{k}] \neq 0, \ \mathbf{k} \in \mathbb{F}_2^n\}$ is chosen uniformly at random over $\mathbb{F}_2^n$;*

- *each non-zero coefficient $\alpha[\mathbf{k}]$ takes values from some set $\mathcal{A}$ according to $\alpha[\mathbf{k}] \sim P_{\mathcal{A}}$ for all $\mathbf{k} \in \mathrm{supp}(\boldsymbol{\alpha})$, and $P_{\mathcal{A}}$ is some probability distribution over $\mathcal{A}$.*

We consider *active learning* under the membership query model. Each query to $f$ at $\mathbf{x} \in \{-1, 1\}^n$ returns the data-label pair $(\mathbf{x}, f(\mathbf{x}) + \varepsilon)$, where $\varepsilon$ is some additive noise. We propose a query framework that leads to a fast reconstruction algorithm, which outputs an estimate $\widehat{\boldsymbol{\alpha}}$ of the polynomial coefficients. The performance of our framework is evaluated by the probability of failing to recover the exact coefficients $\mathbb{P}_F := \Pr(\widehat{\boldsymbol{\alpha}} \neq \boldsymbol{\alpha}) = \mathbb{E}\left[1_{\widehat{\boldsymbol{\alpha}} \neq \boldsymbol{\alpha}}\right]$, where $1_{(\cdot)}$ is the indicator function and the expectation is taken with respect to the noise $\varepsilon$, the randomized construction of our queries, as well as the random polynomial ensemble $\mathcal{F}(s, n, \mathcal{A})$.

## 1.2 Our Approach and Contributions

Particularly relevant to this work are the algorithms on learning decision trees and boolean functions by uncovering the Fourier spectrum of $f$ [3, 5, 10, 12]. Recent papers further show that this problem can be formulated and solved as a compressed sensing problem using random queries [14, 18]. Specifically, [14] gives an algorithm using $O(s^2 n)$ queries based on mutual coherence, whereas the Restricted Isometry Property (RIP) is used in [18] to give a query complexity of $O(sn^4)$. However, this formulation needs to estimate a length-$2^n$ vector and hence the complexity is *exponential* in $n$.

To alleviate the computational burden, [9] proposes a pre-processing scheme to reduce the number of unknowns to $2^s$, which shortens the runtime to $\mathrm{poly}(2^s, n)$ using $O(n2^s)$ samples. However, this method only works with very small $s$ due to the exponential scaling. Under the sparsity regime $s = O(2^{\delta n})$ for some $0 < \delta < 1$, existing algorithms [3, 9, 10, 14, 18], irrespective of using membership queries or random examples, do not immediately apply here because this may require $2^n$ samples (and large runtime) due to the obscured polynomial scaling in $s$.

In our framework, we show that $f$ can be learned exactly in time *almost-linear* in $s$ and *strictly-linear* in $n$, even when the queries are perturbed by random Gaussian noise.

**Theorem 1** (Noisy Learning). *Let $f \in \mathcal{F}(s, n, \mathcal{A})$ where $\mathcal{A}$ is some arbitrarily large but finite set. In the presence of noise $\varepsilon \sim \mathcal{N}(0, \sigma^2)$, our algorithm learns $f$ exactly in terms of the coefficients $\widehat{\boldsymbol{\alpha}} = \boldsymbol{\alpha}$, which runs in time $O(ns \log s)$ using $O(ns)$ queries with probability at least $1 - O(1/s)$.*

The proposed algorithm and proofs are given in the supplementary material. Further, we apply this framework on learning hidden graphs from cut queries. We consider an undirected weighted graph $G = (V, E, W)$ with $|E| = r$ edges and weights $W \in \mathbb{R}^r$, where $V = \{1, \cdots, n\}$ is given but the edge set $E \subseteq V \times V$ is unknown. This generalizes to hypergraphs, where an edge can connect at most $d$ nodes, called the *rank* of the graph. For a $d$-rank hypergraph with $r$ edges, the cut function is a $s$-sparse $d$-bounded pseudo-boolean function (i.e. each monomial depending on at most $d$ variables) where the sparsity is bounded by $s = O(r2^{d-1})$ [9].

On the graph sketching problem, [18] uses $O(sn^4)$ random queries to sketch the sparse temporal changes of a hypergraph in *polynomial* time $\mathrm{poly}(n^d)$. However, [9] shows that it becomes computationally infeasible for small graphs (e.g. $n = 200$ nodes, $r = 3$ edges with $d = 4$), while the LearnGraph algorithm [9] runs in time $O(2^{rd}M + n^2 d \log n)$ using $M = O(2^{rd} d \log n + 2^{2d+1} d^2 (\log n + rd))$ queries. Although this significantly reduces the runtime compared to [14, 18], the algorithm only tackles very sparse graphs due to the scaling $2^r$ and $n^2$. This implies that the sketching needs to be done on relatively small graphs (i.e. $n = 1000$ nodes) over fine sketching intervals (i.e. minutes) to suppress the sparsity (i.e. $r = 10$ within the sketching interval). In this work, we adapt and apply our learning framework to derive an efficient sketching algorithm, whose runtime scales as $O(ds \log s(\log n + \log s))$ by using $O(ds(\log n + \log s))$ queries. We use our adapted algorithm on real datasets and find that we can handle much coarser sketching intervals (e.g. half an hour) and much larger hypergraphs (e.g. $n = 10^5$ nodes).

## 2 Learning Framework

Our learning framework consists of a *query generator* and a *reconstruction engine*. Given the sparsity $s$ and the number of variables $n$, the query generator strategically constructs queries (randomly) and the reconstruction engine recovers the $s$-sparse vector $\boldsymbol{\alpha}$. For notation convenience, we replace each boolean variable $x_i = (-1)^{m[i]}$ with a binary variable $m[i] \in \mathbb{F}_2$ for all $i \in [n]$. Using the notation $\mathbf{m} = [m[1], \cdots, m[n]]^T$ in the Fourier expansion (1), we have

$$u[\mathbf{m}] = \sum_{\mathbf{k} \in \mathbb{F}_2^n} \alpha[\mathbf{k}](-1)^{\langle \mathbf{m}, \mathbf{k} \rangle} + \varepsilon[\mathbf{m}], \tag{2}$$

where $\langle \mathbf{m}, \mathbf{k} \rangle = \oplus_{i \in [n]} m[i] k[i]$ over $\mathbb{F}_2$. Now the coefficients $\alpha[\mathbf{k}]$ can be interpreted as the Walsh-Hadamard Transform (WHT) coefficients of the polynomial $f(\mathbf{x})$ for $\mathbf{x} \in \{-1, 1\}^n$.

## 2.1 Membership Query: A Coding-Theoretic Design

The building block of our *query generator* is the *basic query set* by *subsampling* and *tiny WHTs*:

- **Subsampling**: we choose $B = 2^b$ samples $u[\mathbf{m}]$ indexed selectively by $\mathbf{m} = \mathbf{M}\boldsymbol{\ell} + \mathbf{d}$ for $\boldsymbol{\ell} \in \mathbb{F}_2^b$, where $\mathbf{M} \in \mathbb{F}_2^{n \times b}$ is the *subsampling matrix* and $\mathbf{d} \in \mathbb{F}_2^n$ is the *subsampling offset*.

- **WHT**: a very small $B$-point WHT is performed over the samples $u[\mathbf{M}\boldsymbol{\ell} + \mathbf{d}]$ for $\boldsymbol{\ell} \in \mathbb{F}_2^b$, where each output coefficient can be obtained according to the aliasing property of WHT:

$$U[\boldsymbol{j}] = \sum_{\mathbf{k}:\mathbf{M}^T\mathbf{k}=\boldsymbol{j}} \alpha[\mathbf{k}](-1)^{\langle \mathbf{d}, \mathbf{k} \rangle} + W[\boldsymbol{j}], \quad \boldsymbol{j} \in \mathbb{F}_2^b, \tag{3}$$

where $W[\boldsymbol{j}] = \frac{1}{\sqrt{B}} \sum_{\boldsymbol{\ell} \in \mathbb{F}_2^b} \varepsilon[\mathbf{M}\boldsymbol{\ell} + \mathbf{d}](-1)^{\langle \boldsymbol{\ell}, \boldsymbol{j} \rangle}$ is the observation noise with variance $\sigma^2$.

The $B$-point basic query set (3) implies that each coefficient $U[\boldsymbol{j}]$ is the weighted hash output of $\alpha[\mathbf{k}]$ under the hash function $\mathbf{M}^T\mathbf{k} = \boldsymbol{j}$. From a coding-theoretic perspective, the coefficient $U[\boldsymbol{j}]$ for constitutes a parity constraint of the coefficients $\alpha[\mathbf{k}]$, where $\alpha[\mathbf{k}]$ enters the $\boldsymbol{j}$-th parity if $\mathbf{M}^T\mathbf{k} = \boldsymbol{j}$. If we can induce a set of parity constraints that mimic good error-correcting codes with respect to the unknown coefficients $\alpha[\mathbf{k}]$, the coefficients can be recovered iteratively in the spirit of peeling decoding, similar to that in LDPC codes. Now it boils down to the following questions:

- How to choose the subsampling matrix $\mathbf{M}$ and how to choose the query set size $B$?
- How to recover the coefficients $\alpha[\mathbf{k}]$ from their aliased observations $U[\boldsymbol{j}]$?

In the following, we illustrate the principle of our learning framework through a simple example with $n = 4$ boolean variables and sparsity $s = 4$.

## 2.2 Main Idea: A Simple Example

Suppose that the $s = 4$ non-zero coefficients are $\alpha[0100], \alpha[0110], \alpha[1010]$ and $\alpha[1111]$. We choose $B = s = 4$ and use two patterns $\mathbf{M}_1 = [\mathbf{0}_{2\times2}^T, \mathbf{I}_{2\times2}^T]^T$ and $\mathbf{M}_2 = [\mathbf{I}_{2\times2}^T, \mathbf{0}_{2\times2}^T]^T$ for subsampling, where all queries made using the same pattern $\mathbf{M}_i$ are called a *query group*.

In this example, by enforcing a zero subsampling offset $\mathbf{d} = \mathbf{0}$, we generate only *one* set of queries $\{U_c[\boldsymbol{j}]\}_{\boldsymbol{j} \in \mathbb{F}_2^b}$ under each pattern $\mathbf{M}_c$ according to (3). For example, under pattern $\mathbf{M}_1$, the chosen samples are $u[0000], u[0001], u[0010], u[0011]$. Then, the observations are obtained by a $B$-point WHT coefficients of these chosen samples.

For illustration we assume the queries are noiseless:

$$U_1[00] = \alpha[0000] + \alpha[0100] + \alpha[1000] + \alpha[1100],$$
$$U_1[01] = \alpha[0001] + \alpha[0101] + \alpha[1001] + \alpha[1101],$$
$$U_1[10] = \alpha[0010] + \alpha[0110] + \alpha[1010] + \alpha[1110],$$
$$U_1[11] = \alpha[0011] + \alpha[0111] + \alpha[1011] + \alpha[1111].$$

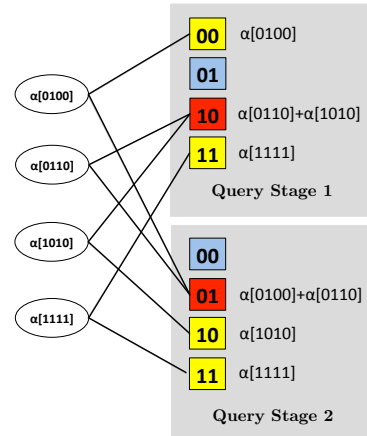

Generally speaking, it is impossible to reconstruct the coefficients from these queries. However, since the coefficients are sparse, then the observations are reduced to

$$U_1[00] = \alpha[0100], \qquad U_2[00] = 0$$
$$U_1[01] = 0, \qquad U_2[01] = \alpha[0100] + \alpha[0110]$$
$$U_1[10] = \alpha[0110] + \alpha[1010], \quad U_2[10] = \alpha[1010]$$
$$U_1[11] = \alpha[1111], \qquad U_2[11] = \alpha[1111].$$

The observations are captured by a *bipartite graph*, which consists of $s = 4$ left nodes and 8 right nodes (see Fig. 2).

Figure 2: Example of a bipartite graph for the observations.

### 2.2.1 Oracle-based Decoding

We illustrate how to decode the unknown $\alpha[\mathbf{k}]$ from the bipartite graph in Fig. 2 with the help of an "oracle", and then introduce how to get rid of this oracle. The right nodes can be categorized as:

- *Zero-ton*: a right node is a zero-ton if it is not connected to any left node.
- *Single-ton*: a right node is a single-ton if it is connected to *only one* left node. We refer to the index $\mathbf{k}$ and its associated value $\alpha[\mathbf{k}]$ as the *index-value pair* $(\mathbf{k}, \alpha[\mathbf{k}])$.
- *Multi-ton*: a right node is a multi-ton if it is connected to *more than one* left node.

The oracle informs the decoder exactly which right nodes are *single-tons* as well as the corresponding index-value pair $(\mathbf{k}, \alpha[\mathbf{k}])$. Then, we can learn the coefficients iteratively as follows:

**Step (1)** select all edges in the bipartite graph with right degree 1 (i.e. detect presence of single-tons and the index-value pairs informed by the oracle);

**Step (2)** remove (peel off) these edges and the left and right end nodes of these single-ton edges;

**Step (3)** remove (peel off) other edges connected to the left nodes that are removed in **Step (2)**;

**Step (4)** remove contributions of the left nodes removed in **Step (3)** from the remaining right nodes.

Finally, decoding is successful if all edges are removed. Clearly, this simple example is only an illustration. In general, if there are $C$ *query groups* associated with the subsampling patterns $\{\mathbf{M}_c\}_{c=1}^C$ and query set size $B$, we define the bipartite graph ensemble below and derive the guidelines for choosing them to guarantee successful peeling-based recovery.

**Definition 2** (Sparse Graph Ensemble). *The bipartite graph ensemble* $\mathcal{G}(s, \eta, C, \{\mathbf{M}_c\}_{c \in [C]})$ *is a collection of $C$-regular bipartite graphs where*

- *there are $s$ left nodes, each associated with a distinct non-zero coefficient $\alpha[\mathbf{k}]$;*

- *there are $C$ groups of right nodes and $B = 2^b = \eta s$ right nodes per group, and each right node is characterized by the observation $U_c[\boldsymbol{j}]$ indexed by $\boldsymbol{j} \in \mathbb{F}_2^b$ in each group;*

- *there exists an edge between left node $\alpha[\mathbf{k}]$ and right node $U_c[\boldsymbol{j}]$ in group $c$ if $\mathbf{M}_c^T \mathbf{k} = \boldsymbol{j}$, and thus each left node has a regular degree $C$.*

Using the construction of $\{\mathbf{M}_c\}_{c=1}^C$ given in the supplemental material, the decoding is successful over the ensemble $\mathcal{G}(s, \eta, C, \{\mathbf{M}_c\}_{c \in [C]})$ if $C$ and $B$ are chosen appropriately. The key idea is to avoid excessive aliasing by exploiting a sufficiently large but finite number of groups $C$ for diversity and maintaining the query set size $B$ on par with the sparsity $O(s)$.

**Lemma 1.** *If we construct our query generator using $C$ query groups with $B = \eta s = 2^b$ for some redundancy parameter $\eta > 0$ satisfying:*

| $C$ | 2 | 3 | 4 | 5 | 6 | $\cdots$ |
|---|---|---|---|---|---|---|
| $\eta$ | 1.0000 | 0.4073 | 0.3237 | 0.2850 | 0.2616 | $\cdots$ |

Table 1: Minimum value for $\eta$ given the number of groups $C$

*then the oracle-based decoder learns $f$ in $O(s)$ peeling iterations with probability $1 - O(1/s)$.*

### 2.2.2 Getting Rid of the Oracle

Now we explain how to detect single-tons and obtain the index-value pair without an oracle. We exploit the diversity of subsampling offsets $\mathbf{d}$ from (3). Let $\mathbf{D}_c \in \mathbb{F}_2^{P \times n}$ be the offset matrix containing $P$ subsampling offsets, where each row is a chosen offset. Denote by $\boldsymbol{U}_c[\boldsymbol{j}] := [\cdots, U_{c,p}[\boldsymbol{j}], \cdots]^T$ the vector of observations (called *observation bin*) associated with the $P$ offsets at the $\boldsymbol{j}$-th right node, we have the general observation model for each right node in the bipartite graph as follows.

**Proposition 1.** *Given the offset matrix $\mathbf{D} \in \mathbb{F}_2^{P \times n}$, we have*

$$\boldsymbol{U}_c[\boldsymbol{j}] = \sum_{\mathbf{k} \,:\, \mathbf{M}_c^T \mathbf{k} = \boldsymbol{j}} \alpha[\mathbf{k}] (-1)^{\mathbf{D}_c \mathbf{k}} + \mathbf{w}_c[\boldsymbol{j}], \tag{4}$$

*where $\mathbf{w}_c[\boldsymbol{j}] \triangleq [\cdots, W_{c,p}[\boldsymbol{j}], \cdots]^T$ contains noise samples with variance $\sigma^2$, $(-1)^{(\cdot)}$ is an element-wise exponentiation operator and $(-1)^{\mathbf{D}_c \mathbf{k}}$ is the offset signature associated with $\alpha[\mathbf{k}]$.*

In the same simple example, we keep the subsampling matrix $\mathbf{M}_1$ and use the set of offsets $\mathbf{d}_0 = [0,0,0,0]^T$, $\mathbf{d}_1 = [1,0,0,0]^T$, $\mathbf{d}_2 = [0,1,0,0]^T$, $\mathbf{d}_3 = [0,0,1,0]^T$ and $\mathbf{d}_4 = [0,0,0,1]^T$ such that $\mathbf{D}_1 = [\mathbf{0}_{1\times4}; \mathbf{I}_4]$. The observation bin associated with the subsampling pattern $\mathbf{M}_1$ is:

$$\boldsymbol{U}_1[\boldsymbol{j}] = [U_{1,0}[\boldsymbol{j}], U_{1,1}[\boldsymbol{j}], U_{1,2}[\boldsymbol{j}], U_{1,3}[\boldsymbol{j}], U_{1,4}[\boldsymbol{j}]]^T. \tag{5}$$

For example, observations $\boldsymbol{U}_1[01]$ and $\boldsymbol{U}_1[10]$ are given as

$$\boldsymbol{U}_1[01] = \alpha[0100] \times \begin{bmatrix} 1 \\ (-1)^0 \\ (-1)^1 \\ (-1)^0 \\ (-1)^0 \end{bmatrix}, \ \boldsymbol{U}_1[10] = \alpha[0110] \times \begin{bmatrix} 1 \\ (-1)^0 \\ (-1)^1 \\ (-1)^1 \\ (-1)^0 \end{bmatrix} + \alpha[1010] \times \begin{bmatrix} 1 \\ (-1)^1 \\ (-1)^0 \\ (-1)^1 \\ (-1)^0 \end{bmatrix}.$$

With these bin observations, one can effectively determine if a check node is a zero-ton, a single-ton or a multi-ton. For example, a single-ton, say $\boldsymbol{U}_1[01]$, satisfies $|U_{1,0}[01]| = |U_{1,1}[01]| = |U_{1,2}[01]| = |U_{1,3}[01]| = |U_{1,4}[01]|$. Then, the index $\mathbf{k} = [k[1], k[2], k[3], k[4]]^T$ and the value of a single-ton can be obtained by a simple ratio test

$$\begin{cases} (-1)^{\widehat{k}[1]} &= \frac{U_{1,1}[01]}{U_{1,0}[01]} = (-1)^0 \\ (-1)^{\widehat{k}[2]} &= \frac{U_{1,2}[01]}{U_{1,0}[01]} = (-1)^1 \\ (-1)^{\widehat{k}[3]} &= \frac{U_{1,3}[01]}{U_{1,0}[01]} = (-1)^0 \\ (-1)^{\widehat{k}[4]} &= \frac{U_{1,4}[01]}{U_{1,0}[01]} = (-1)^0 \end{cases} \implies \begin{cases} \widehat{k}[1] = 0 \\ \widehat{k}[2] = 1 \\ \widehat{k}[3] = 0 \\ \widehat{k}[4] = 0 \\ \widehat{\alpha}[\widehat{\mathbf{k}}] = U_{1,0}[01] \end{cases}$$

The above tests are easy to verify for all observations such that the index-value pair is obtained for peeling. In fact, this detection scheme for obtaining the oracle information is mentioned in the noiseless scenario [16] by using $P = n+1$ offsets. However, this procedure fails in the presence of noise. In the following, we propose the general detection scheme for the noisy scenario while using $P = O(n)$ offsets.

## 3 Learning in the Presence of Noise

In this section, we propose a robust bin detection scheme that identifies the type of each observation bin and estimate the index-value pair $(\mathbf{k}, \alpha[\mathbf{k}])$ of a single-ton in the presence of noise. For convenience, we drop the group index $c$ and the node index $\boldsymbol{j}$ without loss of clarity, because the detection scheme is identical for all nodes from all groups. The bin detection scheme consists of the single-ton detection scheme and the zero-ton/multi-ton detection scheme, as described next.

### 3.1 Single-ton Detection

**Proposition 2.** *Given a single-ton with $(\mathbf{k}, \alpha[\mathbf{k}])$ observed in the presence of noise $\mathcal{N}(0, \sigma^2)$, then by collecting the signs of the observations, we have*

$$\mathbf{c} = \mathbf{D}\mathbf{k} \oplus \mathsf{sgn}\left[\alpha[\mathbf{k}]\right] \oplus \mathbf{z}$$

*where $\mathbf{z}$ contains $P$ independent Bernoulli variables with probability at most $\mathbb{P}_\mathrm{e} = e^{-\eta B \alpha_\mathrm{min}^2/2\sigma^2}$, and the sign function is defined as $\mathsf{sgn}\left[x\right] = 1$ if $x < 0$ and $\mathsf{sgn}\left[x\right] = 0$ if $x > 0$.*

Note that the $P$-bit vector $\mathbf{c}$ is a received codeword of the $n$-bit message $\mathbf{k}$ over a binary symmetric channel (BSC) under an unknown flip $\mathsf{sgn}\left[\alpha[\mathbf{k}]\right]$. Therefore, we can design the offset matrix $\mathbf{D}$ according to linear block codes. The codes should include $\mathbf{1}$ as a valid codeword such that both $\mathbf{D}\mathbf{k}$ and $\mathbf{D}\mathbf{k} \oplus \mathbf{1}$ can be decoded correctly and then obtain the correct codeword $\mathbf{D}\mathbf{k}$ and hence $\mathbf{k}$.

**Definition 3.** *Let the offset matrix $\mathbf{D} \in \mathbb{F}_2^{P\times n}$ constitute a $P \times n$ generator matrix of some linear code, which satisfies a minimum distance $\tilde{\beta}P$ with a code rate $R(\beta) > 0$ and $\beta > \mathbb{P}_\mathrm{e}$.*

Since there are $n$ information bits in the index $\mathbf{k}$, there exists some linear code (i.e. $\mathbf{D}$) with block length $P = n/R(\beta)$ that achieves a minimum distance of $\beta P$, where $R(\beta)$ is the rate of the code [15]. As long as $\beta > \mathbb{P}_\mathrm{e}$, it is obvious that the unknown $\mathbf{k}$ can be decoded with exponentially decaying probability of error. Excellent examples include the class of expander codes or LDPC codes, which admits a linear time decoding algorithm. Therefore, the single-ton detection can be performed in time $O(n)$, same as the noiseless case.

## 3.2 Zero-ton and Multi-ton Detection

The single-ton detection scheme works when the underlying bin is indeed a single-ton. However, it does not work on isolating single-tons from zero-tons and multi-tons. We address this issue by further introducing $P$ extra random offsets.

**Definition 4.** *Let the offset matrix* $\widetilde{\mathbf{D}} \in \mathbb{F}_2^{P \times n}$ *constitute a* $P \times n$ *random matrix consisting of independent identically distributed (i.i.d.) Bernoulli entries with probability* $1/2$.

Denote by $\widetilde{U} = [\widetilde{U}_1, \cdots, \widetilde{U}_P]^T$ the observations associated with $\widetilde{\mathbf{D}}$, we perform the following:

- **zero-ton verification**: the bin is a zero-ton if $\|\widetilde{U}\|^2/P \leq (1+\gamma)\sigma^2/B$ for some $\gamma \in (0, 1)$.
- **multi-ton verification**: the bin is a multi-ton if $\|\widetilde{U} - \widehat{\alpha}[\widehat{\mathbf{k}}](-1)^{\widetilde{\mathbf{D}}\widehat{\mathbf{k}}}\|^2 \geq (1 + \gamma)\sigma^2/B$, where $(\widehat{\mathbf{k}}, \widehat{\alpha}[\widehat{\mathbf{k}}])$ are the single-ton detection estimates.

It is shown in the supplemental material that this bin detection scheme works with probability at least $1 - O(1/s)$. Together with Lemma 1, the learning framework in the presence of noise succeeds with probability at least $1 - O(1/s)$. As detailed in the supplemental material, this leads to a overall sample complexity of $O(sn)$ and runtime of $O(ns \log s)$.

## 4 Application in Hypergraph Sketching

Consider a $d$-rank hypergraph $G = (V, E)$ with $|E| = r$ edges, where $V = \{1, \cdots, n\}$. A cut $S \subseteq V$ is a set of selected vertices, denoted by the boolean cube $\mathbf{x} = [x_1, \cdots, x_n]$ over $\{\pm 1\}^n$, where $x_i = -1$ if $i \in S$ and $x_i = 1$ if $i \notin S$. The value of a specific cut $\mathbf{x}$ can be written as

$$f(\mathbf{x}) = \sum_{e \in E} \left[ 1 - \left( \prod_{i \in e} \frac{(1 + x_i)}{2} + \prod_{i \in e} \frac{(1 - x_i)}{2} \right) \right]. \tag{6}$$

Letting $x_i = (-1)^{m[i]}$, we have $f(\mathbf{x}) = u[\mathbf{m}] = \sum_{\mathbf{k} \in \mathbb{F}_2^n} c[\mathbf{k}](-1)^{\langle \mathbf{k}, \mathbf{m} \rangle}$ with $x_i = (-1)^{m[i]}$ for all $i \in [n]$, where the coefficient $c[\mathbf{k}]$ is a scaled WHT coefficient. Clearly, if the number of hyperedges is small $r \ll 2^n$ and the maximum size of each hyperedge is small $d \ll n$, the coefficients $c[\mathbf{k}]$'s are sparse and the sparsity can be well upper bounded by $s \leq r2^{d-1}$. Now, we can use our learning framework to compute the sparse coefficients $c[\mathbf{k}]$ from only a few cut queries. Note that in the graph sketching problem, the weight of $\mathbf{k}$ is bounded by $d$ due to the special structure of cut function. Therefore, in the noiseless setting, we can leverage the sparsity $d$ and use much fewer offsets $P \ll n$ in the spirit of compressed sensing. In the supplemental material, we adapt our framework to derive the `GraphSketch` bin detection scheme with even lower query costs and runtime.

**Proposition 3.** *The* `GraphSketch` *bin detection scheme uses* $P = O(d(\log n + \log s))$ *offsets and successfully detects single-tons and their index-value pairs with probability at least* $1 - O(1/s)$.

Next, we provide numerical experiments of our learning algorithm for sketching large random hypergraphs as well as actual hypergraphs formed by real datasets[2]. In Fig. 3, we compare the probability of success in sketching hypergraphs with $n = 1000$ nodes over 100 trials against the `LearnGraph` procedure[3] in [9], by randomly generating $r = 1$ to 10 hyperedges with rank $d = 5$. The performance is plotted against the number of edges $r$ and the query complexity of learning. As seen from Fig. 3, the query complexity of our framework is significantly lower ($\leq 1\%$) than that of [9].

### 4.1 Sketching the Yahoo! Messenger User Communication Pattern Dataset

We sketch the hypergraphs extracted from Yahoo! Messenger User Communication Pattern Dataset [19], which records communications for 28 days. The dataset is recorded entry-wise as *(day, time, transmitter, origin-zipcode, receiver, flag)*, where *day* and *time* represent the time stamp of each message, the *transmitter* and *receiver* represent the IDs of the sender and the recipient, the *zipcode* is a spatial stamp of each message, and the *flag* indicates if the recipient is in the contact list. There are $10^5$ unique users and 5649 unique zipcodes. A hidden hypergraph structure is captured as follows.

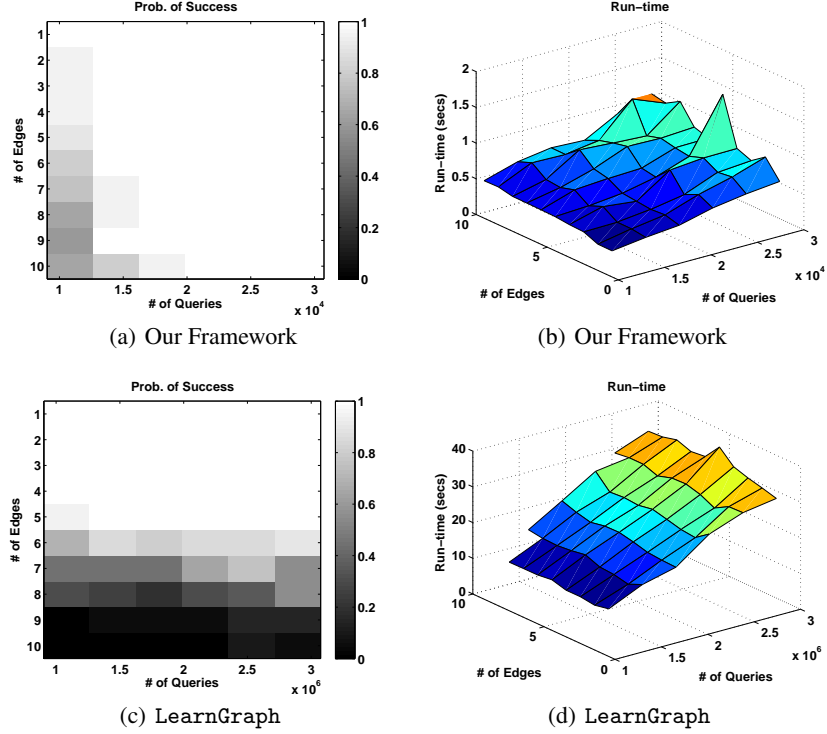

|   |   |
|:-:|:-:|
| (a) Our Framework | (b) Our Framework |
| (c) `LearnGraph` | (d) `LearnGraph` |

Figure 3: Sketching performance of random hypergraphs with $n = 1000$ nodes.

Over an interval $\delta t$, each sender with a unique zipcode forms a hyperedge, and the recipients are the members of the hyperedge. By considering $T$ consecutive intervals $\delta t$ over a set of $\delta z \ll 5649$ zipcodes, the communication pattern gives rise to a hypergraph with only few hyperedges in each interval and each hyperedge contains only few $d$ nodes. The complete set of nodes in the hypergraph $n$ is the set of recipients who are active during the $T$ intervals. In Table 2, we choose the sketching interval $\delta t = 0.5$hr and consider $T = 5$ intervals. For each interval, we extract the communication hypergraph from the dataset by sketching the communications originating from a set of $\delta z = 20$ zipcodes[4] by posing queries constructed at random in our framework. We average our performance over 100 trial runs and obtain the success probability.

| Temporal Graph | $n$ | # of edges ($E$) | degree ($d$) | $1 - \mathbb{P}_F$ | Run-time (sec) |
|:-:|:-:|:-:|:-:|:-:|:-:|
| (9:00 a.m. $\sim$ 9:30 a.m.) | 12648 | 87 | 9 | 0.97 | 422.3 |
| (9:30 a.m. $\sim$ 10:00 a.m.) | 12648 | 102 | 8 | 0.99 | 310.1 |
| (10:00 a.m. $\sim$ 10:30 a.m.) | 12648 | 109 | 7 | 0.99 | 291.4 |
| (10:30 a.m. $\sim$ 11:00 a.m.) | 12648 | 84 | 9 | 0.93 | 571.3 |
| (11:00 a.m. $\sim$ 11:00 a.m.) | 12648 | 89 | 10 | 0.93 | 295.1 |

Table 2: Sketching performance with $C = 8$ groups and $P = 421$ query sets of size $B = 128$.

We maintain $C = 8$ groups of queries with $P = 421$ query sets of size $B = 256$ per group throughout all the experiments (i.e., $8.6 \times 10^5$ queries $\approx 60n$). It is also seen that we can sketch the temporal communication hypergraphs from the real dataset over much larger intervals (0.5 hr) than that by `LearnGraph` (around 30 sec to 5 min), also more reliably in terms of success probability.

## 5   Conclusions

In this paper, we introduce a coding-theoretic active learning framework for sparse polynomials under a much more challenging sparsity regime. The proposed framework effectively lowers the query complexity and especially the computational complexity. Our framework is useful in sketching large hypergraphs, where the queries are obtained by specific graph cuts. We further show via experiments that our learning algorithm performs very well over real datasets compared with existing approaches.

## Footnotes

[1]The notation is defined as $[n] := \{1, \cdots, n\}$.

[2]We used MATLAB on a Macbook Pro with an Intel Core i5 processor at 2.4 GHz and 8 GB RAM.

[3]We would like to acknowledge and thank the authors [9] for providing their source codes.

[4]We did now show the performance of `LearnGraph` because it fails to work on hypergraphs with the number of hyperedges at this scale with a reasonable number of queries (i.e., $\leq 1000n$), as mentioned in [9].

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
