[Supplementary Material]

# Supplemental Material for "An Active Learning Framework using Sparse-Graph Codes for Sparse Polynomials and Graph Sketching"

**Xiao Li**
UC Berkeley
xiaoli@berkeley.edu

**Kannan Ramchandran**
UC Berkeley
kannanr@berkeley.edu

## 1 Proof of Theorem 1

It has been shown in Lemma 1 that the oracle-based peeling decoder succeeds with probability at least $1 - O(1/s)$, therefore we only have to prove that the failure probability of the bin detection routine is upper bounded by $O(1/s)$. Let $E_0$ be the event where the detector makes a mistake in any of the $O(s)$ peeling iteration, then as long as $\Pr(E_0) = O\left(\frac{1}{s}\right)$, we have

$$\mathbb{P}_F = \Pr\left(\text{supp}\left(\widehat{\boldsymbol{\alpha}}\right) \neq \text{supp}\left(\boldsymbol{\alpha}\right) \middle| E_0^c\right) \Pr\left(E_0^c\right) + \Pr\left(\text{supp}\left(\widehat{\boldsymbol{\alpha}}\right) \neq \text{supp}\left(\boldsymbol{\alpha}\right) \middle| E_0\right) \Pr\left(E_0\right)$$
$$\leq \Pr\left(\text{supp}\left(\widehat{\boldsymbol{\alpha}}\right) \neq \text{supp}\left(\boldsymbol{\alpha}\right) \middle| E_0^c\right) + \Pr\left(E_0\right) = O\left(1/s\right).$$

Since the first term in the last inequality is obtained from Lemma 1 for the peeling decoder with an oracle, it remains to show that $\Pr(E_0) = O\left(\frac{1}{s}\right)$ holds, which is shown in details in Appendix C and Appendix D in [1].

Since we have $O(n)$ coded offsets for the single-ton detection, and another $O(n)$ random offsets for the zero-ton/multi-ton detection, we have a total of $O(n)$ offsets. As a result, this leads to a sample cost of $C\eta s \times O(n) = O(sn)$, which is the same as the noiseless case [6]. In terms of complexity, the runtime for each bin detection is $O(n)$, contributing to a total of $O(sn)$ complexity across all $O(s)$ bins. However, this complexity is dominated by the query constructor for generating $O(n)$ sets of $B$-point WHT, each imposing a complexity of $O(s \log s)$. This gives a total complexity of $O(ns \log s)$, same as the noiseless case [6].

## 2 Proof of Lemma 1

Here we first provide a sketch of the proof and provide the details in Appendix B in [1]. The key is to analyze the decoding process through peeling off all the edges in the bipartite graph ensemble $\mathcal{G}(s, \eta, C, \{\mathbf{M}_c\}_{c\in[C]})$. Our analysis is similar to the arguments in [2,5] using the so-called *density evolution* analysis from modern coding theory, which tracks the average density[1] of the remaining edges in the graph at each peeling iteration of the algorithm. Although the proof techniques are similar to those from [2] and [5], the graph used in our peeling decoder is different from those in [2,5]. This leads to fairly important differences in the analysis, such as the degree distributions of the graphs and the expansion properties of the graphs. Since the analysis is quite space demanding and technically driven, we provide a brief outline of the proof elements highlighting the main technical components here.

- **Density evolution**: We analyze the performance of our peeling decoder over a *typical graph* (i.e., cycle-free) of the ensemble $\mathcal{G}(s, \eta, C, \{\mathbf{M}_c\}_{c\in[C]})$ for a fixed number of peeling

iterations $i$. We assume that a local neighborhood of every edge in the graph is cycle-free (tree-like) and derive a recursive equation that represents the average density $p_i$ of remaining edges[2] in the graph at iteration $i$.

$$p_i = \left(1 - e^{-\frac{1}{\eta}p_{i-1}}\right)^{C-1}, i = 1, 2, 3, \cdots \tag{1}$$

Clearly, the probability $p_i$ can be made arbitrarily small for a sufficiently large but finite $i > 0$ as long as $C$ and $\eta$ are chosen properly. One can find the minimum value $\eta$ for a given $C$ to guarantee $p_i < p_{i-1}$, which is shown in Table I in Lemma 1. This suggests that in a finite number of iterations $i$, the remaining edges in the graph is $sCp_i$ on average, which can be made arbitrarily small for a sufficiently large $i$.

- **Convergence to density evolution**: Using a Doob martingale argument as in [5] and [3], we show that the local neighborhood of most edges of a randomly chosen graph from the ensemble $\mathcal{G}(s, \eta, C, \{\mathbf{M}_c\}_{c\in[C]})$ is cycle-free with high probability. This proves that with high probability, our peeling decoder removes all but an arbitrarily small fraction of the left nodes $sp_i$ (i.e., $C$ edges associated with each left node are removed at the same time after being decoded) in $i$ iterations.

- **Graph expansion property**: We show that if the sub-graph consisting of the remaining edges is an "expander", and if our peeling decoder successfully removes all but a sufficiently small fraction of the left nodes from the graph, then it removes all the remaining edges of the graph successfully. As long as the number of groups satisfies $C \geq 3$, we show that our graph ensemble is an expander with high probability. This completes the decoding of all the non-zero coefficients.

Since the graph expansion property in the analysis suggests $C \geq 3$, the minimum redundancy parameter in the query generator is at least $\eta \geq 0.4073$ for $C = 3$ according to Table I in Lemma 1. Finally, the failure probability of the oracle-based peeling decoder is bounded by the sum of the probability of not achieving the density evolution result, and the probability of the remaining graph not being an expander even when the density evolution result is met. Both of these events occur with probability $O(1/s)$, which approaches zero asymptotically in $s$. Last but not least, since there are a total of $O(s)$ edges in the graph, and there is at least one edge being peeled off in each iteration with high probability, the result in Lemma 1 follows.

## 3 `GraphSketch` **Bin Detection Scheme**

The observation vector is denoted by $\boldsymbol{U} = [\cdots, U_p, \cdots]^T$ and the offset matrix is $\mathbf{D} \in \mathbb{F}_2^{P \times n}$. In the absence of noise, we specifically impose a reference offset $\mathbf{d}_0 = \mathbf{0}$ such that the `type` of the observation vector can be easily determined as

$$\text{type} = \begin{cases} \texttt{zero-ton}, & \text{if } |U_p| = 0, \forall p = 0, 1, \cdots, P \\ \texttt{single-ton}, & \text{if } |U_p/U_0| = 1 \\ \texttt{multi-ton}, & \text{if } |U_p/U_0| \neq 1 \end{cases}. \tag{2}$$

This procedure guarantees that the single-tons are identified from zero-tons and multi-tons with probability one as long as the coefficients $\alpha[\mathbf{k}]$ take generic values from some continuous distribution $P_{\mathcal{A}}$ indicated by the polynomial ensemble $\mathcal{F}(s, n, \mathcal{A})$.

After ruling out zero-tons and multi-tons, we focus on the single-ton detection in the `GraphSketch` bin detection scheme. To do so, we use the sign[3] of each $U_p = \alpha[\mathbf{k}](-1)^{\langle \mathbf{d}_p, \mathbf{k} \rangle}$ for $p = 0, \cdots, P$ to find $\mathbf{k}$, because the sign is a linear measurement of the unknown index $\mathsf{sgn}[U_p] = \langle \mathbf{d}_p, \mathbf{k} \rangle \oplus \mathsf{sgn}[\alpha[\mathbf{k}]]$ over $\mathbb{F}_2$. Since the oracle-based decoder succeeds with probability at least $1 - O(1/s)$, it suffices to show that the error probability of the single-ton detection is upper bounded by $O(1/s)$.

### 3.1 Design of Subsampling Offsets for `GraphSketch`

Recall from Proposition 3 that the sign vector of the observations of a single-ton is

$$\mathsf{sgn}[\boldsymbol{U}] = \mathbf{c} = \mathbf{D}\mathbf{k} \tag{3}$$

over $\mathbb{F}_2$. Without loss of generality, we assume that the sign $\operatorname{sgn}[\alpha[\mathbf{k}]]$ is $0$ since it can be obtained by simply introducing a reference offset $\mathbf{d}_0 = \mathbf{0}$ as mentioned in Section 2.2.2. Therefore, if the offset matrix is chosen as $\mathbf{D} = \mathbf{I}_{n \times n}$, the index can be directly read out as $\mathbf{c} = \mathbf{k}$, as shown in [6].

In graph sketching, each index $\mathbf{k}$ has a sparsity $d$. Therefore, we can leverage the sparsity of $\mathbf{k}$ and use much fewer offsets $P \ll n$ in the spirit of compressed sensing.

To facilitate the description of our subsampling offset design for graph sketching, we introduce the the row-tensor product $\boxtimes$:

$$
\begin{bmatrix} a & b & c & d \\ e & f & g & h \end{bmatrix} \boxtimes \begin{bmatrix} 0 & 0 & 1 & 1 \\ 0 & 1 & 0 & 1 \end{bmatrix} = \begin{bmatrix} 0 & 0 & c & d \\ 0 & b & 0 & d \\ 0 & 0 & g & h \\ 0 & f & 0 & h \end{bmatrix}. \tag{4}
$$

**Definition 1.** *Let $P = P_1 P_2$ with $P_1 = \lambda_1 d$ and $P_2 = 2 \log_2 n + \lambda_2 \log s$ for some $\lambda_1, \lambda_2 > 0$. Given a coding matrix $\mathbf{H} \in \mathbb{F}_2^{P_1 \times n}$ and an index identification matrix $\mathbf{S} \in \mathbb{F}_2^{P_2 \times n}$, the offset matrix is chosen as $\mathbf{D} = \mathbf{H} \boxtimes \mathbf{S}$, where the coding matrix $\mathbf{H}$ and index identification matrix $\mathbf{S}$ are specified below:*

- *Let $\mathbf{H}$ be the $\lambda_1 d \times n$ adjacency matrix of an expander graph with $n$ left nodes and $\lambda_1 d$ right nodes for some universal constant $\lambda_1 > 0$, where any subset of $v$ left nodes (for any $v \leq d$) are connected to at least $v/2$ nodes;*

- *The index identification matrix*

$$
\mathbf{S} := \begin{bmatrix} \mathbf{B} \\ \mathbf{R} \end{bmatrix} \tag{5}
$$

*is a concatenated matrix where $\mathbf{R} = [\mathbf{r}_0, \cdots, \mathbf{r}_{n-1}]$ is a $\lambda_2 \log s \times n$ i.i.d. random Bernoulli(1/2) matrix and $\mathbf{B} = [\mathbf{b}_0, \cdots, \mathbf{b}_{n-1}]$ is the $2 \log_2 n \times n$ augmented binary expansion matrix of $0$ through $n - 1$. For instance, when $n = 4$, $\mathbf{B}$ consists of a binary expansion matrix of $0$ to $3$ and its binary complement:*

$$
\mathbf{B} = \begin{bmatrix} 0 & 0 & 1 & 1 \\ 0 & 1 & 0 & 1 \\ 1 & 1 & 0 & 0 \\ 1 & 0 & 1 & 0 \end{bmatrix}. \tag{6}
$$

### 3.2 Algorithm for Single-ton Detection for `GraphSketch`

With this offset design based on the row-tensor product, the associated sign vector $\mathbf{c}$ can be divided into multiple *bins* $\mathbf{c} = [\cdots, \mathbf{c}_i^T, \cdots]^T$ for $i = 1, \cdots, \lambda_1 d$, where each bin $\mathbf{c}_i$ is

$$
\mathbf{c}_i = \mathbf{S} \operatorname{diag}[\mathbf{h}_i] \mathbf{k}, \quad j = 1, \cdots, \lambda_1 d
$$

where $\mathbf{h}_i$ is the $i$-th row of $\mathbf{H}$. The relationship between the bins $\{\mathbf{c}_i\}_{j=1}^{\lambda_1 d}$ and the non-zero bits in $\mathbf{k}$ is similarly captured by the bipartite graph given by $\mathbf{H}$.

Therefore, the non-zero bit in the $n$-tuple $\mathbf{k}$ can be decoded in a similar peeling fashion. This subroutine is explained as follows. Given $\mathbf{S} = [\mathbf{B}; \mathbf{R}]$, we separate $\mathbf{c}_i$ into two parts as

$$
\mathbf{c}_{i,1} = \mathbf{B} \operatorname{diag}[\mathbf{h}_i] \mathbf{k} \tag{7}
$$

$$
\mathbf{c}_{i,2} = \mathbf{R} \operatorname{diag}[\mathbf{h}_i] \mathbf{k}. \tag{8}
$$

We decode the unknown index $\mathbf{k} = [k[1], \cdots, k[n]]$ bit-by-bit using the following iterative subroutine for each $\mathbf{c}_i$ over $i = 1, \cdots, \lambda_1 d$. At each iteration of this sub-routine, we perform

- *zero-ton test*: $\mathbf{c}_i$ is a zero-ton if $\|\mathbf{c}_i\|^2 = 0$;

- *single-ton search*: assuming that $\mathbf{c}_i$ is a single-ton (i.e. contributed by only one non-zero bit in $\mathbf{k}$), then the estimate $\widehat{q}$ of the location of the non-zero bit is obtained by reading out $\mathbf{c}_{i,1}$ as the binary expansion of the column in $\mathbf{B}$;

- *single-ton test*: if $\mathbf{c}_{i,2}$ matches with the pattern $\mathbf{c}_{i,2} = \mathbf{r}_{\widehat{q}}$ of the $\widehat{q}$-th column of $\mathbf{R}$, then $\mathbf{c}_i$ is indeed a single-ton and the non-zero bit is confirmed as $\widehat{k}[\widehat{q}] = 1$;
- *iterative peeling*: this single-ton contribution can be peeled off from other bins $\mathbf{c}_i$:

$$\mathbf{c}_{i,1} \leftarrow \mathbf{c}_{i,1} \oplus \mathbf{b}_{\widehat{q}}, \quad \mathbf{c}_{i,2} \leftarrow \mathbf{c}_{i,2} \oplus \mathbf{r}_{\widehat{q}}, \tag{9}$$

where $\mathbf{b}_{\widehat{q}}$ is the binary expansion of $\widehat{q}$ and $\mathbf{r}_{\widehat{q}}$ is the $\widehat{q}$-th column of $\mathbf{R}$.

Since $\mathbf{H}$ is an expander such that any subset of $v$ left nodes (for any $v \le d$) are connected to at least $v/2$ nodes, the peeling is guaranteed to recover all unknown left nodes (i.e. all the non-zero bits in $\mathbf{k}$). The construction of the expander graph $\mathbf{H}$ can be obtained offline with high probability [4] and used for all instances of graph sketching. Therefore, the probability of error of this procedure is upper bounded by the probability of confusing a multi-ton with a single-ton, which decays exponentially with respect to $O(\log s)$ given by the tail bounds of $O(\log s)$ random i.i.d. entries in each column of $\mathbf{R}$. Finally, this leads to a probability of error at most $O(1/s)$.

Therefore, if the graph ensemble $\mathcal{G}(s, \eta, C, \{\mathbf{M}_c\}_{c \in [C]})$ guarantees oracle-based peeling, then the GraphSketch bin detection scheme leads to an overall sketching algorithm that uses only $C \times \eta s \times P = O(ds(\log n + \log s))$ queries and runs in time $O(P \times s \log s) = O(ds \log s(\log n + \log s))$.

## Footnotes

[1]The density here refers to fraction of the remaining edges, or namely, the number of remaining edges divided by the total number of edges in the graph.

[2]$p_i$ is the fraction of edges remaining in the graph over $sC$ edges in the $i$-th iteration.

[3]Note that the sign function here is defined as $\mathsf{sgn}[x] = 1$ if $x < 0$ and $\mathsf{sgn}[x] = 0$ if $x > 0$.