[Reviews · NeurIPS 2015]

Submitted by Assigned_Reviewer_1

++++++++++++ Summary:

The paper proposes a new active-learning algorithm to learn a Fourier-sparse function defined over the Boolean cube. Their algorithm exhibiting sublinear sample- and time-complexity. The authors apply their algorithm to the problem of hypergraph sketching and obtain favorable empirical results.

Pros:

- The results are technically sound, and represent significant theoretical improvements over existing methods for learning sparse polynomials (modulo extra assumptions). - The techniques are an interesting synthesis of ideas from sparse recovery, coding theory, and statistical learning theory.

Cons: - The presentation and flow of ideas can be improved. Assumptions, notation, and parameters are not clearly specified in several places. - The approach relies on randomness of the input, and this inhibits wider applicability. - Experimental results are evocative, but somewhat limited.

++++++++++++ Detailed:

- The paper proposes a new algorithm to learn an unknown function (defined over the Boolean cube) that is s-sparse in the Fourier domain from a small number of queries. The algorithm is both sample- and time-efficient. Specifically, when the Fourier support is uniformly chosen at random and the coefficients are from a discrete set, the algorithm requires merely O(ns) samples and can do the reconstruction in O(ns log s) time with high probability. Moreover, if each monomial only depends on d variables, then the algorithm requires merely O(ds logn log s) samples and O(ds log n log^2 s) time.

- The theoretical contributions of the paper are compelling. At first glance, they seemed to represent a significant improvement over existing results for sparse polynomial learning, especially in terms of reconstruction time, where the best known results [9,16,21] were all exponential in either s or n. The catch is that the theory heavily relies upon the random support assumption, which the previous approaches do not make. Moreover, the previous approaches work with queries evaluated over random points in the Boolean cube, while the current approach makes certain actively chosen queries. Despite these caveats, I think the contributions are important in their own right.

- On the negative side, some of the main ideas are obscured by lack of clarity in the presentation. Perhaps this was caused by the need to fit within the page limit. I list some additional comments/suggestions below for improvement. The algorithmic techniques are interesting, but not terribly new -- they seem to follow from a fruitful use of the sublinear sparse Fourier transform approaches of [11,17] and others. Also, it is unclear if the randomness assumption on the unknown function is valid in actual applications (e.g. in graph sketching). The real-data results in Section 3.1 seem to indicate that the ideas are useful, but the authors do not compare with any other existing approaches.

Additional comments: - I believe that the approach of [21] only requires O(sn polylog s) samples, as opposed to O(s n^4) samples, as stated here; e.g. by invoking the better sample-complexity bounds for Fourier-RIP by Cheragchi-Guruswami-Velinger. - Consider highlighting these connections to [11,17] more clearly in the introduction.

- Unclear what 'Proposition' 1 is trying to claim; consider renaming to 'Definition' or 'Remark'. - Can the authors explain where the values of \eta and \lambda_1 in Tables 1 and 2 come from?

- Figure 2 seems strange; for a fixed sparsity level, why doesn't the performance of LearnGraph improve with more queries? - What is P_F in Table 3?

-Consider including comparisons with other existing approaches (e.g. the graph sketching approach in [21] can be conceivably implemented in a reasonable amount of time).
Summary: The paper proposes a new active-learning algorithm, for learning a sparse polynomial over the Boolean cube, that exhibits sublinear sample- and time-complexity. While the presentation of ideas admits room for improvement, the theoretical and experimental results are novel, compelling, and can be of potential significant impact.

Submitted by Assigned_Reviewer_2

Summary: The authors consider the problem of learning n-variate s sparse polynomials 'f'(sparse in the number of monomials) over the boolean field. They consider the average setting, where the polynomial is generated uniformly at random. They use tools from coding theory to derive an algorithm that actively queries f (at O(ns) locations) and learns it in time nearly linear in n,s. Moreover this also holds: (i) in the high sparsity regime where s is allowed to scale exponentially with n and, (ii) in the noisy setting where the queries are corrupted with gaussian noise. They also extend the analysis to the setting where the number of variables in a monomial is bounded and provide experiments on real data sets.

Quality: The results obtained are strong -- the proposed algorithm achieves significant improvements both in terms of sample complexity bounds, and the run time.

Clarity: The paper is overall written decently -- the problem is well motivated and the relation to existing work along with the improvements made is demonstrated clearly. Some comments:

(a) In the line following Eq. (3), the index j is present only in l.h.s and not in r.h.s

(b) In Prop. 1, second line: shouldn't : "generates P a query sets of size B .." be "generates P query sets, each of size B .." ? It would be good to explain how the values in Table 1 are obtained and also comment on the redundancy parameter \eta.

(c) It might be a good idea to add a block diagram that illustrates the decoding process.

Originality: The analysis in the paper appears to be quite novel.

Significance: The results appear to be strong improvements over existing ones -- although the average case setting is considered here as opposed to the worst case setting in previous

works.

Summary: The authors provide improved sample complexity as well as run time bounds for learning sparse polynomials over the boolean field. The results appear to be strong.

Submitted by Assigned_Reviewer_3

The paper studies the problem of actively learning a function f:{-1,1}^n -> R which is a sum of at most s parity functions or in other words its Fourier spectrum is sparse. The algorithm is allowed to query the value of the function at any desired point z in F2^n.

The main result is: For any function whose parity functions are randomly uniformly drawn from the set of all parities and whose coefficients are drawn from a distribution on a bounded set, the algorithm runs in time O(ns logs) with O(ns) queries and learns the function. This is also possible when the queries are noisy.

Another special case of the result is when the parities have at most d variables in them. In this case, the algorithm has O(sd logs logn) query complexity and O(ds log2s logn) run time. Further, this is applied to the problem of learning hypergraphs from cuts (graph sketching) and when the hyperedges are of constant size, the algorithm learns the graph in O*(r) time where O*() involves some poly log terms in n and r. This is strictly better than any algorithm known all of which needs poly in n running time. The results have been compared to another set of algorithms developed for the case of sparse function learning from random samples. Simulations on real data sets have been provided that demonstrate the gain from other algorithms.

Main Ideas:

Actually, in finding coefficients of parities and in also finding the parities, this work designs queries in such a way that a "sparse parity-check like equations" akin to decoding LDPC codes in coding theory is formed and peeling decoders are used to find the solution. In detail the work uses the following two ideas:

Idea 1: Fourier expansion of f is a Walsh-Hadamard Transform with the coefficients of parities serving as the WHT coefficients. Now if one takes the s- point WHT (WHT is also its inverse) of a bunch of queries evaluated on a hyperplane H in F2n of dimension b = O(log s),

then each of the coefficients of the s-point WHT is a very sparse linear combination of only a subset of coefficients of parities. The s-point WHT of the queries and the coefficients of the actual parities form a sparse bipartite graph where peeling decoding is employed. However, this can reveal only the coefficients of the parities but not the parities themselves. For that authors employ the following idea

Idea 2: Each of the s-point WHT of the queries is formed by linearly combining +1s and -1s and the sign depends only on the shift of the hyperplane H from origin and the parity.

So multiple hyperplanes are used to create different +-1 linear combinations. Now these can be used at the time of peeling, to relate the actual parity and the sign of the s-point WHTs for different hyperplanes.

So O(n) hyperplanes are necessary in general. But if the parities are of bounded degree d, then clearly clever design of the hyperplane shifts can result in another compresses sensing problem with bi-partite graphs and peeling decoding which will reduce the number of hyper planes requires to O(d log n).

Strengths: The results for bounded degree parities and its subsequent application graph sketching are novel and they almost match the information theoretic lower bound of O*(r) query complexity for the graph sketching problem. The simulations are very interesting and its comparison with previous algorithm for the random examples case illustrates its utility.

Weaknesses: 1) This is my most important comment/suggestion:

In section 1.2 and in several places, the solution of this work is compared with ref16, ref21 and ref9. But those algorithms are for random samples with no query access. However, the Query access model is a lot more powerful. So I think the authors could try comparing it with Kushilevitz-Mansour algorithm (refer to Thm 3.37 in the Book:Analysis of Boolean functions by O'Donnell). Kushilevitz-Mansour algorithm (with some very simple modifications to the case when f:{-1,1}n-> R) can be made to run in time poly(n,s) and it also designs queries adaptively. Closely, looking at it, it seems like it has very similar queries and run time complexities although I could be wrong. However, the bounded degree case seems novel and the resulting graph sketching application seems novel. But even in this case, the algorithm given in page 85, of the same book, could be modified to start with buckets of size nd instead of size 2n as in conventional Kushilevitz-Mansour algorithm and I think that would give the O(d log n) scaling that accompanies the results in the paper.

However, I am not very sure about the second suggestion. Authors could clarify this point.

2) Minor comments:

a) Choices for matrices Mc in page (4) is not mentioned in the main paper. It seems like it should not matter much. But I am not sure what role it plays in the peeling decoding analysis.

b) Proposition 1 "...where each group generates P a query set.. "-> 'a' should be removed. c) g_{\hat{k}} in algorithm 1 is not defined in the main paper. d) Last but the second line before Definition 3: "that is is ..." -> only one 'is'. e) \mathbf{H} is defined using MATLAB notation of concatenating matrices row wise. Could clarify that notation somewhere. f) Section 3 Line 4 - "..cut xcan be..." - > needs to change.
Summary: The paper contains several nice and novel ideas, borrowing from the LDPC peeling decoding to learn sparse Boolean functions. The peeling decoder is simply the back-substitution step of Gaussian elimination, i.e. a fast way of solving linear equations when the equations have a good structure. This paper designs queries for the boolean function in a clever way to enforce such a structure.

Overall I would recommend acceptance.

However, weakness point 1 is important and should be addressed.

Author Feedback
Author rebuttal: We would like to thank the reviewers for their careful reviews and comments.

Reviewer_1:
i) Regarding novelty compared with [11][17]: our results were inspired by [11][17] but most of the algorithmic aspects (esp. noise robustness, graph sketching) are vastly different from existing works. As suggested by the reviewer, we will highlight the connections to these methods in the introduction.

ii) Regarding the uniform randomness of the unknown function: the random assumption is only needed for our theory but not critical in practice. There is also a fair amount of work in compressed sensing based on random support assumption. The proposed learning framework works very well through numerous experiments we ran. We believe it's the first step to understand the problem and it is an interesting future direction to remove this assumption.

iii) Regarding comparisons with other approaches: in fact, the work [12] suggested by the reviewer has been already compared with LearnGraph in [9] for networks with up to n=1000 nodes, where the LearnGraph algorithm showed much better performance (esp. run-time). Therefore, due to lack of space, we only compared with LearnGraph in Fig. 2 for the same setting n=1000, followed by a large-scale experiment in Table 3 with n=12648 which LearnGraph fails to tackle.

iv) Thanks for pointing to the Fourier-RIP result O(s polylog(s) n). We did not imply that the O(sn^4) is the theoretic minimum, but rather meant to state the result verbatim as in [12]. We will add a remark about this. Also, when s=O(N^\delta) as studied in our paper, since $N=2^n$, the polylog(s) factor is equivalent to poly(n) in the scaling.

v) Regarding Proposition 1: Proposition 1 is about establishing the guidelines for choosing the value \eta given the number of query groups "C". Simply put, it is a phase transition threshold determined by the number of query groups "C" (also refer to our reply to the next question below), above which the algorithm succeeds with high probability. We will make it clearer in the context.

vi) Regarding \eta and \lambda_1 in Table 1 and 2: these values are obtained by studying the density evolution of the proposed peeling decoder, which was attached as supplemental materials in our submission.

vii) Regarding Figure 2: some performance improvement can be observed as the # of queries increase for LearnGraph. For the few points when #edges > 6 where the performance seems a bit unstable, it is due to the relatively small number of trials (100 trials) to average out the inconsistency, because the success probability of LearnGraph is becoming rather small for graphs with #edges > 6.

viii) PF is the probability of failing to recover all the coefficients PF = Pr( \hat{alpha} \neq alpha), which was defined after Definition 1.

Reviewer_2
a) The index j is also on the right hand side in the summation formula, as the summing condition M^T k = j below the summation sign. We will clarify in the paper and make it more explicit.

b) Regarding the values of eta and lambda_1: we refer to our reply (v) to Reviewer_1.

c) Great suggestion. We will add a block diagram for decoding.

Reviewer_3
(1) Regarding weakness point 1: indeed the KM-algorithm shares a similar query model, but the sample complexity and run-time are very different. The KM-algorithm has a sample complexity and run-time poly(s,n), while our algorithm has a provable O(sn) sample complexity and O(n s log s) run-time that is linear in the number of variables n. Furthermore, all our queries are not adaptively designed, but rather obtained and solved iteratively in one shot (not adaptive or sequential). For the d-bounded case, our algorithm requires only O(d s log s log n) samples and O(d s log^2 s log n) run-time. In the graph sketching experiment with n=12648, this makes a big difference.

(2) Regarding weakness point 2: for the KM-algorithm, assume that changing the bucket size may bring down the O(n) factor in sample complexity to (d log n). However, the bottleneck is in the run-time for estimating the Fourier coefficient location k, which is O(n). It is not obvious how KM-algorithm can be adapted to achieve O(d log n) run-time given the shrunk bucket size.

(3) The choices for M_c are stated in the supplemental materials, but generally speaking the theory goes through as long as all the M_c for c=1,..., C span the entire F2^n space. We will clarify this further in the main text.

(4) minor comments (b)-(f). Thanks for carefully reading our paper, we will correct these typos and improve the clarity.

Reviewer_4
We thank the reviewer for his/her kind comments.

Reviewer_5
We thank the reviewer for his/her comments. We will improve on our presentation and delivery in the final version, and hope that the comments from the other reviewers and our response addresses the reviewer's concerns.

Reviewer_6
We thank the reviewer for his/her kind comments.